# Recurrent Memory Transformer

**Aydar Bulatov**[1]
bulatov.as@phystech.edu

**Yuri Kuratov**[1,2]
yurii.kuratov@phystech.edu

**Mikhail S. Burtsev**[1,2]
burtcev.ms@mipt.ru

[1]Neural Networks and Deep Learning Lab,
Moscow Institute of Physics and Technology, Dolgoprudny, Russia
[2]AIRI, Moscow, Russia

## Abstract

Transformer-based models show their effectiveness across multiple domains and tasks. The self-attention allows to combine information from all sequence elements into context-aware representations. However, global and local information has to be stored mostly in the same element-wise representations. Moreover, the length of an input sequence is limited by quadratic computational complexity of self-attention. In this work, we propose and study a memory-augmented segment-level recurrent Transformer (RMT). Memory allows to store and process local and global information as well as to pass information between segments of the long sequence with the help of recurrence. We implement a memory mechanism with no changes to Transformer model by adding special memory tokens to the input or output sequence. Then the model is trained to control both memory operations and sequence representations processing. Results of experiments show that RMT performs on par with the Transformer-XL on language modeling for smaller memory sizes and outperforms it for tasks that require longer sequence processing. We show that adding memory tokens to Tr-XL is able to improve its performance. This makes Recurrent Memory Transformer a promising architecture for applications that require learning of long-term dependencies and general purpose in memory processing, such as algorithmic tasks and reasoning.

## 1   Introduction

Transformers (Vaswani et al., 2017) have been widely adopted across multiple domains and tasks (Radford et al., 2018; Dong et al., 2018; Devlin et al., 2019; Dosovitskiy et al., 2021; Ramesh et al., 2021; Jaegle et al., 2021). The key component of Transformer layer is a self-attention. Self-attention allows to update each sequence element representation with information from all other elements in the sequence. As a result, rich contextual representation for every element is generated at the end of encoding. This way, global sequence-level and local information are stored in a single representation. However, this mixing of two types of information in a single representation has limitations. Distributed storage of global

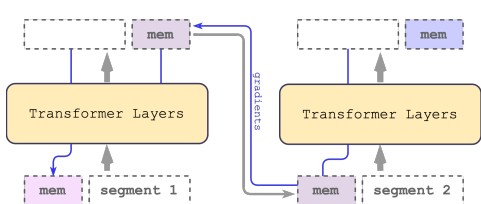

Figure 1: **Recurrent Memory Transformer.** Memory is added as tokens to the input sequence and memory output is passed to the next segment. During training gradients flow from the current segment through memory to the previous segment.

features across all sequence elements results in global features "blurring" and makes it harder to access them. Another well-known deficiency of Transformers is poor scaling of self-attention with

36th Conference on Neural Information Processing Systems (NeurIPS 2022).

input sequence length that hurts its applications to long inputs (Child et al., 2019; Guo et al., 2019; Dai et al., 2019; Beltagy et al., 2020; Ainslie et al., 2020; Zaheer et al., 2020; Wang et al., 2020; Choromanski et al., 2020).

Our work introduces a memory-augmented segment-level recurrent Transformer named Recurrent Memory Transformer (RMT). RMT uses a memory mechanism based on special memory tokens (Burtsev et al., 2020) added to the input sequence. Memory tokens provide additional reserved capacity to the model that could be used to process information which is not directly representing any element in the input sequence. To process long sequences, we split them into segments and pass memory states from a previous to a current segment. This memory passing makes the model recurrent and removes the input sequence length limitations. RMT model can theoretically work with infinite lengths but, in practice, it is limited by memory capacity and the efficiency of memory access/update operations. Our implementation of both memory and recurrence in RMT requires no changes to the Transformer model because modifications are made only to the input and output sequences of the model.

We tested RMT on the tasks that require global information about the whole input sequence to be solved. We use copy, reverse, and associative retrieval tasks in the setting where the input sequence is split into segments. RMT and Transformer-XL perfectly solve these tasks, but exceeding some value of sequence length, RMT starts to outperform Transformer-XL. Also, we experimentally show that the proposed Recurrent Memory Transformer requires less memory size to perform closely to Transformer-XL on language modeling tasks. RMT code and experiments are available[1].

**Contributions**

1. In this study we augment Transformer with token based memory storage and segment-level recurrence.

2. We experimentally evaluate proposed architecture as well as vanilla Transformer and Transformer-XL on memory-intensive tasks such as copy, reverse, associative retrieval, and language modeling. We show that RMT outperforms Transformer-XL for sequence processing tasks and on par with Transformer-XL on language modeling but requires less memory.

3. We show that Tr-XL cache could be combined with RMT leading to better performance on language modeling.

4. We analysed how the Transformer model learns to use memory. Specific interpretable memory read-write patterns of attention are shown.

## 2 Related work

In our study we add a memory to general purpose attention based neural architecture. Memory is a recurrent topic in neural networks research. It had started from the early works (McCulloch and Pitts, 1943; Stephen, 1956) and significantly progressed in 90's with introduction of *Backpropagation Through Time* learning algorithm (Werbos, 1990) and *Long-Short Term Memory* (LSTM) (Hochreiter and Schmidhuber, 1997) neural architecture. Today memory-augmented neural networks (MANNs) usually rely on some kind of recurrent external-memory which is separate from the model's parameters. *Neural Turing Machines* (NTMs) (Graves et al., 2014) and *Memory Networks* (Weston et al., 2014) are equipped with a storage for vector representations that can be accessed with an attention mechanism. Memory Networks (Weston et al., 2014; Sukhbaatar et al., 2015) were designed to enable reasoning by sequential attention over to the content of a memory. NTMs followed by *Differentiable Neural Computer* (DNC) (Graves et al., 2016) and *Sparse DNC* (Rae et al., 2016) are implemented as recurrent neural networks able to write to memory storage over time. All these models are differentiable and can be trained via backpropagation through time (BPTT). Parallel line of research extends recurrent neural networks such as LSTM with data structures like stacks, lists, or queues (Joulin and Mikolov, 2015; Grefenstette et al., 2015). MANN architectures with a more advanced addressing mechanisms such as address-content separation and multi-step addressing were proposed in (Gulcehre et al., 2016, 2017; Meng and Rumshisky, 2018). The Global Context Layer model (Meng and Rumshisky, 2018) uses the idea of address-content separation to solve the difficulty of training content-based addressing in the canonical NTM.

---

[1] `https://github.com/booydar/LM-RMT`. The code, results of the raw experiments and hyperparameters are provided in the supplementary materials and on GitHub.

The recent rise of Transformer models also resulted in introduction of a number of new memory architectures. *Transformer-XL* (Dai et al., 2019) introduces a segment-level recurrence at the level of hidden representations. These representations of a sequence are computed and stored in the cache to be reused as an extended context for the next segment. *Compressive Transformer* (Rae et al., 2019) adds the second layer of memory to Transformer-XL. This memory compresses and stores information from the cache. $\infty$-*former* (Martins et al., 2021) utilizes continuous-space attention and represents input sequence as a continuous signal to make long-term memory unbounded. *Memory Layers* (Lample et al., 2019) model has a product key memory layer instead of a feed-forward layer within Transformer block to increase model capacity.

In many variations of Transformer different sorts of global representations are added. Among them are *Star-Transformer* (Guo et al., 2019), *Longformer* (Beltagy et al., 2020), *GMAT* (Gupta and Berant, 2020), *Extended Transformer Construction* (ETC) (Ainslie et al., 2020) and *Big Bird* (Zaheer et al., 2020). All these architectures re-design self-attention mechanism to reduce it computational complexity with and ensure input coverage with the help of global representations. *Memory Transformer* (Burtsev et al., 2020) keeps Transformer model intact and adds memory by extending input sequence with special memory tokens. Perceiver IO (Jaegle et al., 2021) maps an entire arbitrary input to the fixed number of latent representations. Transformer layers do further processing over latent memory representations only.

Segment-level recurrence in Transformers is actively explored in a number of studies. Transformer-XL, Compressive Transformer keep previous states and re-use them in subsequent segments. Ernie-Doc (Ding et al., 2021) improves processing by using same-layer recurrence instead of attending to previous layer outputs of a precedent segment. Memformer (Wu et al., 2020) introduces a dedicated memory module to keep previous hidden states in summarized representations. Memformer uses two special layers added to the Transformer model. Memory cross-attention layer reads from memory and memory slot attention layer updates it. MART (Lei et al., 2020) has a similar approach as Memformer but uses memory update rules analogous to LSTM (Hochreiter and Schmidhuber, 1997) and GRU (Cho et al., 2014). FeedBack Transformer (Fan et al., 2020) goes further with full, and not segment-level, recurrence. FeedBack Memory merges past hidden representations from all layers into a single vector and makes it accessible to the computations at any layer. The disadvantage of full recurrence is that it is less parallelizable. FeedBack Memory requires every sequence element to be processed sequentially. In segment-level recurrent models, all elements of a segment are processed by Transformer layers in parallel. Only segments are processed sequentially. Staircase Transformer (Ju et al., 2021) combines segment-level recurrence and depth recurrence. Staircase models use the output for previous segments and pass them as input for the next segment. Our Recurrent Memory Transformer is based on special memory tokens similar to Memory Transformer, segment-level recurrence as in Transformer-XL, and depth-recurrent mechanism for memory processing similar to Staircase.

## 3 Recurrent Memory Transformer

Transformer-XL (Dai et al., 2019) extends Transformer model with state re-use cache mechanism for segment-level recurrence and relative position encoding. Input sequence is split on segments processed sequentially. Hidden states computed for the previous segment $M^n$ are cached for each transformer layer $n$. The input of the layer $n$ consists of the last $m$ states from the cached memory and output of previous Transformer layer for the current segment $\tau$:

$$\tilde{H}_\tau^{n-1} = [SG(M_{-m:}^{n-1}) \circ H_\tau^{n-1}],$$

here, SG stands for stop-gradient, $\circ$ denotes concatenation. Cached states allow to increase effective context size of Transformer model and save on compute operations.

Then, $\tilde{H}_\tau^{n-1}$ goes to Transformer layer $TL$ to produce layer $n$ outputs for segment $\tau$:

$$H_\tau^n = TL(Q_\tau^n, K_\tau^n, V_\tau^n), Q_\tau^n = W_q^n H_\tau^{n-1}; K_\tau^n = W_k^n \tilde{H}_\tau^{n-1}, V_\tau^n = W_v^n \tilde{H}_\tau^{n-1}.$$

In Transformer-XL, self-attention layers are modified to use relative position encodings to improve generalization to longer attention lengths. The overall architecture is shown in the Figure 2.

Memory augmented Transformers such as GMAT, ETC, Memory Transformer (Gupta and Berant, 2020; Ainslie et al., 2020; Burtsev et al., 2020) proposed to use special global tokens as storage

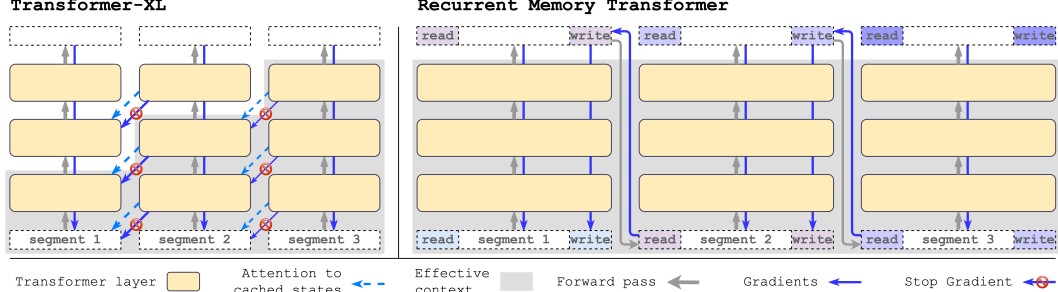

Figure 2: **Comparison of Recurrent Memory Transformer (RMT) and Transformer-XL architectures.** Recurrent Memory Transformer augments Transformer with global memory tokens and passes them to allow a segment-level recurrence. Special read/write memory tokens are added to the input sequence. Multiple memory tokens can be used in each read/write block. Updated representations of write memory are passed to the next segment. During training, RMT uses BPTT to propagate gradient to previous segments through memory tokens representation. Effective context length for recurrence with memory is not limited by the depth of a network which is the case for the cache of Transformer-XL.

for representations. Usually, memory tokens are added to the beginning of the input sequence. However, in decoder-only architectures the causal attention mask makes impossible for memory tokens at the start of the sequence to collect information from the subsequent tokens. On the other hand, if memory tokens are placed at the end of the sequence then preceding tokens unable to access their representations. To solve this problem we add a recurrence to the sequence processing. Representations of memory tokens placed at the end of the segment are used as an input memory representations at the start as well as at the end of the next segment.

In the Recurrent Memory Transformer input is augmented with special [mem] tokens, processed in a standard way along with the sequence of tokens. Each memory token is a real-valued vector. $m$ memory tokens are added at the beginning of the segment tokens representations $H_\tau^0$ and the same $m$ tokens are added at the end:

$$\tilde{H}_\tau^0 = [H_\tau^{mem} \circ H_\tau^0 \circ H_\tau^{mem}], \bar{H}_\tau^N = \text{Transformer}(\tilde{H}_\tau^0), [H_\tau^{read} \circ H_\tau^N \circ H_\tau^{write}] := \bar{H}_\tau^N,$$

here $N$ is a number of Transformer layers.

The starting group of memory tokens functions as a read memory that allows sequence tokens to attend to memory states produced at the previous segment. The ending group works as a write memory that can attend to all current segment tokens and update representation stored in the memory. As a result, $H_\tau^{write}$ contains updated memory tokens for the segment $\tau$.

Segments of the input sequence are processed sequentially. To enable recurrent connection between segments, we pass outputs of the memory tokens from the current segment to the input of the next segment:

$$H_{\tau+1}^{mem} := H_\tau^{write}, \tilde{H}_{\tau+1}^0 = [H_{\tau+1}^{mem} \circ H_{\tau+1}^0 \circ H_{\tau+1}^{mem}].$$

Both memory and recurrence in the RMT are based only on global memory tokens. It allows to keep the backbone Transformer unchanged and make RMT memory augmentation compatible with any model from the Transformer family. Memory tokens operate only on the input and output of the model. In this study we implement RMT on top of the original Transformer-XL code. Both architectures are shown in Figure 2.

Recurrence in the RMT is different compared to the Transformer-XL because the former stores only $m$ memory vectors per segment. On the other hand, the Transformer-XL stores $m \times N$ vectors per segment. Also, in the RMT model memory representations from the previous segment are processed by Transformer layers together with the current segment tokens. This makes memory part of RMT effectively deeper in a number of applied Transformer layers $\tau \times N$. Additionally, we allow all memory tokens in the read/write block to access all other tokens in the same block. The causal attention mask is applied only to tokens of the input sequence (Figure 6(d)).

We train the RMT with Backpropagation Through Time (BPTT). During backward pass, unlike in Transformer-XL, memory gradients are not stopped between segments. The number of previous segments to backpropagate is a hyperparameter of a training procedure. We vary BPTT unroll in our experiments from 0 to 4 previous segments. Increasing this parameter is computationally expensive

and requires a lot of GPU RAM. However, techniques such as gradient checkpointing could be used to alleviate this problem.

# 4   Experiments

We designed our experiments to evaluate the ability of Recurrent Memory Transformers to preserve long-term dependencies across multiple input segments. The first set of experiments includes copy, reverse, associative retrieval, and quadratic equations tasks. The second one addresses language modeling task for word-level on WikiText-103 (Merity et al., 2017) and for character-level on enwik8 (Mahoney, 2006). We compare Recurrent Memory Transformer with Transformer and Transformer-XL models.

Our RMT implementation is based on Transformer-XL repository[2]. The full set of hyperparameters is available in our repository as well as in supplementary materials. Language modeling experiments follow the same model and training hyperparameters as Transformer-XL. WikiText-103 experiments use 16-layer Transformers (10 heads, 410 hidden size, 2100 intermediate FF), enwik8 – 12 layer Transformers (8 heads, 512 hidden size, 2048 intermediate FF). We used Adam optimizer Kingma and Ba (2015) with linear schedule learning rate starting from $0.00025$ for 200,000 steps for WikiText-103 and 400,000 steps for enwik8. We refer to Transformer-XL with memory size equal to zero as a Baseline. With this experimental setup we were able to reproduce results for the Transformer-XL model close to the original paper.

**Algorithmic Tasks.** We evaluate RMT on algorithmic tasks that require information about the whole input sequence to be solved successfully. In a recurrent setting, the model has to keep information about all previous segments to make predictions.

In the *Copy* task, an input sequence should be replicated twice after a special start-to-generate token. In the *Reverse* task, an input sequence should be generated in a reverse order. Input for the *Associative Retrieval* task consists of $N$ key-value pairs. Then one key is randomly selected, and the task is to produce an appropriate value for the selected key. Another task is to solve quadratic equations. One example consists of an equation, its solution with discriminant, and an answer. The task is to generate a solution and answer, while only answer quality is evaluated.

For all tasks, input and output sequences are split into segments and processed by models sequentially. Datasets for algorithmic tasks were randomly pre-generated, the same data was used in all experiments, and character-level tokenization was used. Because Transformer-XL and RMT are decoder-only Transformer models, we don't compute loss over the input sequence before the start-to-generate token. The loss is computed over target sequence segments only (see Appendix A.1 for details).

**Language Modeling and NLP.** We use two standard benchmarks for language modeling: WikiText-103 and enwik8. WikiText-103 (Merity et al., 2017) is used for word-level language modeling and contains 103M words from English Wikipedia articles. Enwik8 (Mahoney, 2006) is used for character-level and consists of $10^8$ first bytes of XML text dump of the English Wikipedia. Vocabulary contains 267735 words and 204 characters for Wikitext-103 and enwik8 tokenizers accordingly.

We compare Recurrent Memory Transformer with decoder-only Transformer and Transformer-XL as baselines. Model size and training parameters are selected to match Transformer-XL paper. For Wikitext-103 an input context length was set to 150 tokens, and for enwik8 it was set to 512 characters. Another set of experiments inspected how RMT handles long-term dependencies and recurrence. We increased the number of segments and recurrent steps by making segments smaller (50 tokens for WikiText-103, 128 characters for enwik8). The increased number of recurrent steps makes language modeling tasks harder for RMT because information has to be stored in the same amount of memory for more steps.

As a testbed for the real-life application scenario we select popular long-text classification benchmark Hyperpartisan news (Kiesel et al., 2019). Instead of pre-training RMT from scratch we add recurrent memory mechanism to the most widely adopted models from HuggingFace Transformers (Wolf et al., 2020). Specifically, we augment 500 input tokens of already pretrained BERT-base, RoBERTa-base, DeBERTa-base and T5-base with the recurrent memory of size 10 and fine-tune on the target task.

---

[2]`https://github.com/kimiyoung/transformer-xl`

# 5 Results

Baseline, Transformer-XL (Tr-XL) and RMT perform perfectly in the single segment setting on copy and reverse tasks (Figure 3). In this case, the models do not need recurrence because the whole sequence is available. When the number of segments is larger than one, non-recurrent baseline struggles to solve tasks, but both memory models demonstrate ability to retain required information from the previous segments in memory.

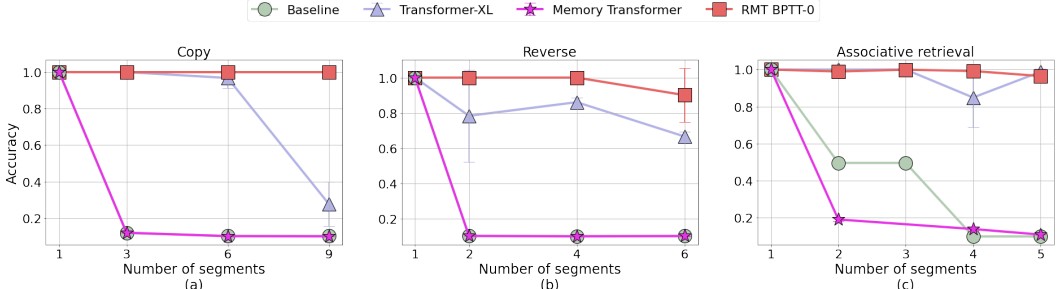

Figure 3: **RMT outperforms Transformer-XL on Copy and Reverse tasks as a number of segments increases.** Panels show test set per-character accuracy on copy, reverse, and associative retrieval tasks (from left to right). Memory/cache size equals to the length of a segment for both models. RMT does not pass gradients between segments in this experiment. MT results are the same as for the Baseline. Source/target sequence lengths for copy, reverse, and associative retrieval tasks: 24/48, 24/24, 10/1.

On Copy and Reverse tasks as a number of segments increases, RMT starts to outperform Transformer-XL with memory sizes less than the number of all previous tokens. With the number of segments up to 6 mean accuracy of Transformer-XL drops by up to 0.2 points, and with 9 segments plunges close to the baseline without memory. Associative Retrieval results are similar with the number of segments up to 4. RMT manages to solve the task with Transformer-XL closely behind. However, in the setting with 5 segments, RMT performance slightly decreases and Transformer-XL average accuracy rises higher.

We analyze how a number of segments, sequence length, a length of training context, and memory size affect models' performance on Copy task (Figure 4). As we split a sequence into more segments it becomes more crucial to be able to pass information between segments. We split 360 tokens of source + target sequence into multiple segments. In Figure 4a we observe that Transformer-XL performance starts to degrade and eventually falls to the baseline model performance as the number of segments increases. In contrast, RMT continues to solve the task perfectly. In a more extreme setting, when we keep memory size fixed, but increase the total length of a sequence to copy Transformer-XL fails shortly, while RMT starts to gradually degrade only after the length of 720 tokens (Figure 4b).

On the Quadratic Equations task (Table 1) we have checked that it is possible to solve the task with the Transformer baseline and no segmentation used. The baseline in this case defines upper bound for this task. With multiple segments recurrency RMT solves the task perfectly, while Transformer-XL finds the task challenging.

The results of experiments on word-level language modeling on WikiText-103 are shown in Table 2. In the first section with a segment length of 150, Tr-XL and RMT outperform the baseline and Memory Transformer (MemTr) by a large margin. It shows the significance of increased effective context length by Tr-XL cache or RMT memory for language modeling. RMT improves over MemTr memory mechanism with read/write blocks. The best RMT models with

Table 1: **Quadratic equations task.** Sequence of 180 tokens consists of quadratic equation, a solution, and an answer. It is split into a number of segments with an answer in the last segment. Accuracy equals 1.0 if the full answer is predicted correctly.

| MODEL | MEMORY | SEGMENTS | $\text{ACC}_{\pm \text{STD}}$ |
|---|---|---|---|
| BASELINE | 0 | 1 | $0.99 \pm 0.01$ |
| TRANSFORMER-XL | 30 | 6 | $0.93 \pm 0.02$ |
| RMT | 30 | 6 | $0.99 \pm 0.002$ |

memory size 10 and 25 show similar performance as Transformer-XL with a memory size equal to 75. RMT learns to use smaller memory more effectively than Transformer-XL. Additionally, the smaller memory size of RMT leads to reducing required GPU memory for running the model.

To force models to process longer recurrent dependencies the size of a segment is set to 50, so the number of recurrent steps increases. RMT with memory size 1 shows similar results to Transformer-

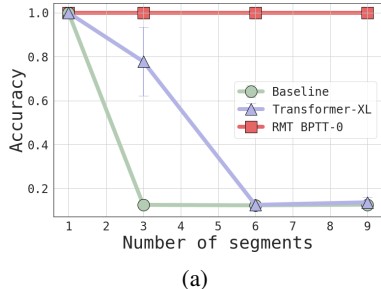 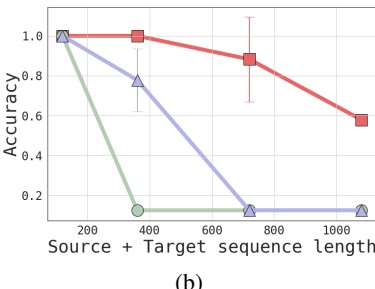

|(a)|(b)|

Figure 4: **RMT scales better with a number of segments and sequence size.** (a) RMT is able to solve copy task perfectly up to 9 segments for a fixed sequence length of 360, while Tr-XL fails. (b) RMT learns to use memory of the same fixed size (60 tokens) more effectively than TR-XL as a sequence length to copy increases (a segment size is 120 for the both models).

XL with memory size 10. It is worth noting that Transformer-XL memory consists of hidden representations from all layers (in this case, it is $10 \times 16$ vectors) when RMT memory is only `memory_size` vectors. Transformer-XL with memory size 50 and RMT with memory size 5 show similar perplexity values (see Appendix A.5).

RMT could be combined with Tr-XL cache. In this case Tr-XL cache could be seen as short-term memory keeping the nearest context and RMT memory as long-term memory. Such combination leads to the best results on WikiText-103 improving over Tr-XL.

Table 2: **Language modeling on WikiText-103.** Average perplexity for the best performed variations of RMT models reported (see full results in Appendix A.5). Underlined values show Tr-XL and RMT models with close results. RMT models with smaller memory sizes achieve similar scores to Tr-XL models with larger memory. Combination of cache with recurrent memory (Tr-XL + RMT) shows the best performance.

| MODEL | MEMORY | SEGMENT LEN | PPL $\pm_{\mathrm{STD}}$ |
|---|---|---|---|
| TR-XL (PAPER) | 150 | 150 | 24.0 |
| BASELINE | 0 | 150 | $29.95 \pm 0.15$ |
| MEMTR | 10 | 150 | $29.63 \pm 0.06$ |
| TR-XL (OURS) | 150 | 150 | $24.12 \pm 0.05$ |
| TR-XL | 25 | 150 | $25.57 \pm 0.02$ |
| TR-XL | 75 | 150 | $\underline{24.68} \pm 0.01$ |
| RMT BPTT-3 | 10 | 150 | $25.04 \pm 0.07$ |
| RMT BPTT-2 | 25 | 150 | $\underline{24.85} \pm 0.31$ |
| TR-XL + RMT | 75+5 | 150 | $24.47 \pm 0.05$ |
| TR-XL + RMT | 150+10 | 150 | $\mathbf{23.99} \pm 0.09$ |
| BASELINE | 0 | 50 | $39.05 \pm 0.01$ |
| TR-XL | 100 | 50 | $\mathbf{25.66} \pm 0.01$ |
| TR-XL | 50 | 50 | $\underline{26.54} \pm 0.01$ |
| TR-XL | 25 | 50 | $27.57 \pm 0.09$ |
| TR-XL | 10 | 50 | $28.98 \pm 0.11$ |
| RMT BPTT-1 | 1 | 50 | $\underline{28.71} \pm 0.03$ |
| RMT BPTT-3 | 10 | 50 | $\underline{26.37} \pm 0.01$ |

On enwik8 RMT models with memory size 5 and Transformer-XL with memory size 40 show similar results. Confirming that RMT learns to use smaller amounts of memory representation more effectively. All results for enwik8 dataset are shown in Appendix A.4.

Recurrent Memory Transformer learns to make predictions depending on `#BPTT_unrolls` over previous segments $+1$ current segment. Transformer-XL does not use BPTT and relies only on `memory_size` cached states and current segment making in total: `memory_size` + `segment_length` tokens. In Figure 5a, we compare RMT and Tr-XL according to the described value of visible context at training time.

RMT with a single memory vector could be trained to achieve lower perplexity as Transformer-XL with memory size 10. This means that RMT can learn to compress information from the previous observations better. Another observation is that RMT with memory sizes 10 and 25 performs only a bit weaker compared to Transformer-XL even when Transformer-XL has access to more non-compressed states (50, 100, 200) from previous segments. In general, training RMT with unrolling gradients in earlier segments drastically improves scores thus showing the importance of BPTT training but, we observe instabilities and out-of-memory issues during RMT training for a larger memory sizes with deeper BPTT unrolls.

RMT wins a lot when only one memory token is added but then the effect from increasing memory size from 5 to 50 fades (Figure 5b). Still, RMT with memory size 5 have performance on par with Transformer-XL with cache 50, confirming that RMT learns to store more compact representations. The results suggest that there is some optimal memory size for RMT to solve the task, and further increase does not add much.

Proposed recurrent memory mechanism affects only input and gradient flows of the augmented core model. This might be an important advantage because the memory can be added to already pretrained

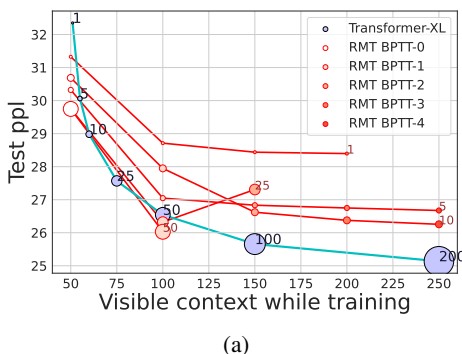
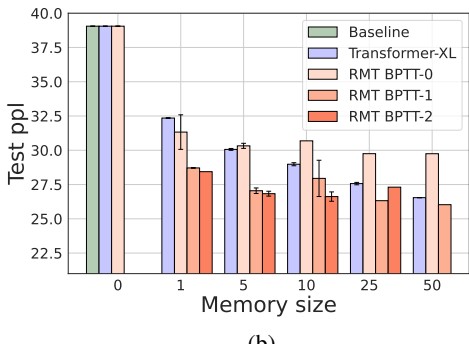

(a)              (b)

Figure 5: **Deeper BPPT unrolling improves RMT scores on WikiText-103** (a) Visible context at training time can be increased by deeper BPTT unrolls for RMT or enlarging cache for Tr-XL. Larger visible context leads to lower perplexity for both models (marker size corresponds to memory size). (b) Recurrence improves performance of RMT compared to Tr-XL for the same memory sizes.

model. Evaluation results for four memory augmented language models fine tuned for long text classification are presented in the Table 3. Incorporation of 10 memory tokens in the input sequence of 512 allows to encode longer stretches of a text up to 2000 tokens and significantly improve metrics for the majority of models. Moreover, a combination of recurrent memory with RoBERTa-base results in state of the art performance for the Hyperpartisan news classification task (Kiesel et al., 2019). Interestingly, many competing models have input size of 4096 that is at least twice longer compared to RMT extended counterparts but still lag behind.

To get an understanding of memory operations, learned by RMT for algorithmic tasks we visualise attention maps for copy and reverse tasks (Figure 6). In each RMT attention map sequence tokens are preceded by read memory, located at the top left corner, and followed by write memory at the bottom right. Diagonal at the central part of the fig.6(a) (top) shows classic attention of token sequence

Table 3: **Hyperpartisan news detection.** Models starting with RMT are taken from HuggingFace Transformers and augmented with 10 memory tokens and recurrence before fine-tuning. Train/valid/test split as in (Beltagy et al., 2020) and metric is F1.

| MODEL [INPUT SIZE] | NUMBER OF SEGMENTS | | | |
|---|---|---|---|---|
| | 1 | 2 | 3 | 4 |
| BIG BIRD [4096] (ZAHEER ET AL., 2020) | 92.20 | | | |
| LONGFORMER [4096] (BELTAGY ET AL., 2020) | 94.80 | | | |
| GRAPH-ROBERTA [512x100] (XU ET AL., 2021) | 96.15 | | | |
| ERNIE-DOC-LARGE [640] (DING ET AL., 2021) | 96.60 | | | |
| ERNIE-SPARSE [4096] (LIU ET AL., 2022) | 92.81 | | | |
| RMT BERT-BASE-CASE [512] | 91.60 | 94.12 | 93.06 | 94.34 |
| RMT ROBERTA-BASE [512] | 94.87 | **97.20** | **96.72** | **98.11** |
| RMT DEBERTA-V3-BASE [512] | 94.17 | 96.78 | 94.80 | 94.80 |
| RMT T5-BASE [512] | **94.99** | 95.32 | 96.12 | 97.20 |

to itself, but the bottom diagonal represents the operation of writing of sequence tokens to memory in straight order. When completing reverse (fig.6(a) bottom) the model learns to write the sequence to the memory in the reversed order, which is in line with common sense.

When it comes to reproducing the target sequence, the model accesses memory (fig.6(b)) and writes to the output sequence. Another operation (fig.6(c)) is rewriting from read memory to write memory. It is commonly used by RMT in settings with larger number of segments to keep information about recent segments longer.

Transformer-XL mechanism of accessing memory (fig.6(d)) does not allow straightforward writing to memory without changing sequence token representations. Sequential reading from cache is represented by diagonals on Transformer-XL attention maps. Using token representations as storage harms model performance in tasks with larger number of segments. For reverse task with 4 segments Transformer-XL with limited memory size 6 (Appendix B Figure 9(b)) attempts to mix representations of tokens and read multiple symbols from one cached state in the next segments giving average accuracy of 0.8 on the target task. Despite having the same memory size, RMT manages to compress the whole segment in memory tokens (Appendix B Figure 9(a)) and achieve mean accuracy 1.

Visualizations from Figure 6 and Appendix B Figure 9 provide evidence to support our hypotheses that Tr-XL has to mix representations from previous and current segments in the same hidden states to pass information between segments. Also, visualizations show how memory tokens in RMT help

mitigate such kind of mixing. RMT ability of sequence compression to memory is illustrated in Appendix A.1 Figure 8. For copy with 6 segments RMT compresses and then reads the sequence of 12 tokens with just 6 memory tokens. For Transformer-XL decreasing memory size harms the accuracy score significantly with number of segments larger than 2.

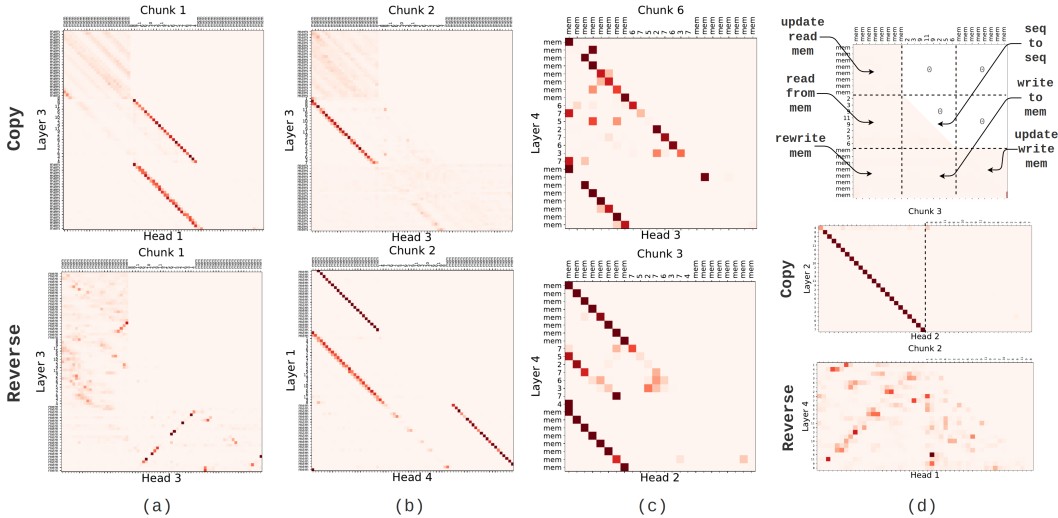

Figure 6: **Selected attention map patterns of memory models.** (color intensity corresponds to attention score) RMT with segment length=24, memory size=24 (a) write to memory, (b) read from memory. (c) RMT, segment length=8, memory size=8, rewrite from read memory to write memory. (d) Transformer-XL, segment length=24, memory size=24 read from the previous hidden states.

## 6 Conclusions

In this paper we introduced Recurrent Memory Transformer a simple recurrent memory augmentation of Transformer model. RMT is implemented by extension of an input sequence with special global memory tokens and segment-level recurrence. Importantly, our method allows to learn more compact sequence representations and improve existing pretrained models without extensive additional compute, thus making practical machine learning applications more energy efficient and environmentally friendly.

In our experiments we compared RMT with Transformer baseline and Transformer-XL which is a well-known modification of Transformer for long sequences. RMT almost perfectly solves Copy, Reverse as well as quadratic equations tasks for sequences consisting of multiple segments outperforming Transformer-XL. It also demonstrates quality for associative retrieval task on par with Transformer-XL. As expected, baseline Transformer fails to solve these tasks for multi-segment settings.

RMT trained as a language model performs significantly ahead of Transformer baseline and shows quality metrics similar to Transformer-XL but for up to 10 times smaller memory size. Experimental results demonstrate that for fixed memory size backpropagating gradients for more segments improves performance of RMT. Proposed approach to memory augmentation is quite universal and might be easily applied to any pretrained transformer based model as demonstrated by achievement of state of the art results for long text classification task by fine tuning a combination of RoBERTa and RMT.

Analysis of attention maps suggests that better RMT performance can be related to more effective storage of input representations in dedicated memory tokens compared to mixing representations storage in Transformer-XL. RMT could be combined with Transformer-XL cache and improve the performance of both models.

Overall, results of the study show that dedicated memory storage and recurrence provided by Recurrent Memory Transformer make it a promising architecture for applications that require learning of long-term dependencies and general purpose in-memory processing, such as algorithmic tasks and reasoning. Furthermore, we believe that RMT could open the way for adding memory and recurrence to other models in the Transformer family.

## Acknowledgments and Disclosure of Funding

This work was supported by a grant for research centers in the field of artificial intelligence, provided by the Analytical Center for the Government of the Russian Federation in accordance with the subsidy agreement (agreement identifier 000000D730321P5Q0002) and the agreement with the Moscow Institute of Physics and Technology dated November 1, 2021 No. 70-2021-00138.

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
