# OpenReview forum: "Recurrent Memory Transformer"
_NeurIPS.cc/2022/Conference — NeurIPS 2022 Accept_

### Official Review · Reviewer_hj1Y · 2022-07-09

**Rating:** 6
**Confidence:** 4
**Soundness:** 3 good
**Presentation:** 3 good
**Contribution:** 3 good

**Summary:**

This paper shows a new method for adding recurrence to Transformers via special overlapping special memory tokens between segments, hence, the new Transformer architecture is called Recurrent Memory Transformer (RMT). A special type of tokens, aka memory tokens are prepended and appended to each segment, and the number M for adding how many memory tokens is a new hyperparameter. The gradients can propagate to past segments through memory tokens, which is the distinct feature from previous architecture such as Transformer XL. The paper uses Transformer XL as the baseline.

**Questions:**

Most of the details about the Transformer architecture and its training procedure are not given in the paper. The width of the Transformer used for the experiments is not given at all. You can't just link to the open-source library, and explain that you used the same configuration defined there, for the following reasons: 1) The GitHub repository may not exist in the future. 2) The configs can change by the owners of the library. 3) It's not a recommended way to communicate experimental details. Although, you would like to contain more interesting results from your findings, but the details about the model architecture, optimization, data processing (tokenization), training procedure are important for reproducibility. And we should all care about reproducibility.

I think the expressiveness of memory doesn't solely depend on the number of memory tokens (M steps), but also by the depth and the width of the Transformer. Did the authors ablate these as well?

What are the solutions in practice to tackle training instability and OOM issue? Does this mean that, for large-scale models, TrXL would be more advantageous than RMT?

**Ethics Review Area:**

["I don’t know"]

**Limitations:**

The paper didn't properly discussed about the limitations of the proposed method rather than adding only one sentence in Section 5. "Furthermore, we observed instabilities and out-of-memory issues during RMT training for a larger number of BPTT unrolls and memory sizes". I suggest being more clear about this and how it can affect the scaling property of the proposed method.

The negative societal impact was not discussed in the paper, so N/A.

**Strengths And Weaknesses:**

Transformer is easy to perform parallel training, and it suffers less from vanishing gradients compared to recurrent neural networks (RNNs), which made it as popular nowadays. However, Transformer is more expensive than RNNs due to its extensive use of attention mechanism, especially during inference. Another downside is increasing the sequence length modelled by Transformers is difficult as the state-space of the memory is much larger than RNNs. And again, the computational cost for attending to previous context increase linearly as the context length increases.

Previous works have addressed this issue by caching the Transformer state of immediate previous segment and inventing a new type of positional embedding that only cares about relational difference, or compressing the states of previous context. The former (TrXL) has been often considered as the baseline for Transformers introducing recurrence. Although, TrXL can arrange longer-term contexts for sequence modelling tasks, there are discontinuities in the backward pass so the model doesn't effectively model long-term temporal dependencies. Also, the Transformer state of previous segment is directly given as the memory, which can increase the computational cost and memory requirement for the forward pass. This works provides a way to 1) pass the gradients from the future segments, which is presumably more principled way of modelling long-term dependencies, 2) connect consecutive segments with small overheads (M memory tokens, where M is much smaller size than the segment size, N).

I think the claim "Recurrence with memory has no limitations on effective context length" written in the caption of Figure 2 doesn't sound right. It is difficult to increase the context length without trading off other factors such as the size of model parameters due to limited memory space of the hardwares. Which is mentioned in the Section 5, the authors wrote that RMT can suffer from instabilities and OOM issues when using larger number of BPTT unrolls. This could affect the scaling property of RMT.

Overall, the writing is good, it conveys the idea very clearly. Their experiments look convincing, which I will leave my comments below.

Comments about the experiments:
They results from synthetic tasks looks good (Figure 4). However, the performance of the RMT on WikiText-103 shows that TrXL still performs the best, and RMT can achieve the performance similar to TrXL variant which uses half-sized memory. I think the choice of segment length (only 50 and 150) is quite limited, and also the length of memory in this particular experiment. And 150 tokens sounds quite short as a sequence length in in the first place. It would be nice to add extra longer length such as 300, 600 for this task as well. However, the experiments on enwiki8 (Figure 5) show nicely that the baseline Transformer and TrXL both shows worse results as the sequence length increases, but RMT is more robust and preserves its performance.

It would be nice if you can write more details about the models, optimizer, tokenization and training procedure. These details should be contained in the main manuscript for reproducibility and readability.

One thing that I would like to propose to the authors as a reviewer to make the paper to look more stronger, is to add a bit more baselines that tried to compress the previous contexts, e.g., compressive Transformer. Also, RMT and Feedback Transformer has a close relationship, the RMT can be seen as block-level recurrent version of feedback Transformer. One could argue that instead of providing one memory token per each step, RMT provides M memory tokens as the context for processing a segment of N tokens. Maybe adding Feedback Transformer and its variants (providing M memory state similar to RMT for processing arbitrary number of tokens in the new segment).

Minor comment:
Figure 7 referenced in the main paper is not included in the main, but in the appendix. It would be better to make this more clear

---

> ### Author Response · Authors · 2022-08-01
> **Response to Reviewer hj1Y**
>
> Q1.  We seriously take your concerns about reproducibility and have added more details about the models, optimizer, tokenization and training procedure to the updated version of the paper. We also provide a full set of hyperparameters used in each of experimental runs in supplementary materials. Hopefully, provided details will be sufficient for successful replication of our work. We also plan to publish implementation of our method as an extension for Hugging Face framework to enable easy memory augmentation for models published on the platform.
>
> Q2. We have not yet accomplished a detailed study how depth and width of the core model would affect efficiency of memory encoding. We plan to address these questions in our upcoming experiments. As well we have plans to implement a memory with a separate “controller” transformer subnetwork with its own parameters because operations with memory might require different transformations compared to processing of representations of sequence elements.
>
> Q3. Right now we observe instability of training for memory sizes larger than 10. To address these instabilities we are actively exploring  addition of axillary training losses, gradient clipping and other regularization techniques. At the same time training problems are partially softened by two factors: (1) experimental results show that even for the modest number of memory tokens ensuring stable training we still get pretty significant improvement  even for BPPT unrolls up to 4 (Fig. 5(a)); and (2) RMT can be combined with a memory cache of Transformer-XL making it applicable to large scale models.
>
> Thank you for pointing to similarities between RMT and Feedback Transformer, although we know the work but never seen it under this angle. We will explore connections between these two models more deeply in our future work.
>
> We agree that the claim "Recurrence with memory has no limitations on effective context length" is not correct. We have changed Figure 2 caption to better convey the idea intended to be expressed as “Effective context length for recurrence with memory is not limited by the depth of a network which is the case for the cache of Transformer-XL.”
>
> We also properly referenced all figures that come from an Appendix in the main text.

---

### Official Review · Reviewer_VBgN · 2022-07-10

**Rating:** 6
**Confidence:** 4
**Soundness:** 4 excellent
**Presentation:** 4 excellent
**Contribution:** 2 fair

**Summary:**

In this paper, authors propose a recurrent Transformer including the memory to store and process local and global information in a input segment and transfer info between segments. The add special tokens at the start and end of each input segment and define the output of these special tokens as READ and WRITE. The representation of the WRITE will be fed to the next segment. They claim that their model can achieve similar results with Transformer-XL with smaller smaller memory sizes.

**Questions:**

Please refer the above section.

**Limitations:**

Yes.

**Strengths And Weaknesses:**

Strengths:
1. The idea is clever and easy to follow. To resolve the issues brought by the autoregressive property, READ token can be the memory for its following tokens and WRITE token can be the summary of this segment which can be used for the next segments.
2. The experiment is sufficient and comparison to the Transformer-XL is also sufficient to me.

Weakness:
1. My main concern is the training process of the RMT. To me, since the gradient will be back propagated to the very beginning, is there any forgetting for the memory token? So the token in the last segment will forget the contextual information of the first segment.
2. Since Transformer-XL will stop the gradient for each segment, I think the training process will be more stable than RMT. Can you confirm this?
3. Since you claim that RMT can benefit the longer sequence modeling due to the quadratic computational complexity, it would be reasonable to include some other efficient transformer and its alternatives, such as sparse transformer, Luna, S4 in the Table 2 for comparison.

---

> ### Author Response · Authors · 2022-08-01
> **Response to Reviewer VBgN**
>
> Response to Q1
>
> We have not explicitly tested how signal fades during propagation from the first segment to the last. At least in theory gradient is able to flow as long as we unfold BPTT. Results presented on the fig. 5a show that unrolling for 4 segments is still better compared to unrolling for 3 segments. This points to the conclusion that the signal propagates at least over 4 segments.
>
> Response to Q2
>
> Yes, in the Section 5 we mention that “we observed instabilities and out-of-memory issues during RMT training for a larger number of BPTT unrolls and memory sizes”. We continue to experiment with axillary training losses, gradient clipping and other techniques to stabilize training.
>
> Response to Q3
>
> To address this question we included in the updated version of the paper comparison of popular transformer models such as BERT,  RoBERTa, DeBERTa, T5 augmented with the proposed recurrent memory mechanism vs. efficient variations of transformer such as  BigBird, Longformer, Linformer, Graph-roberta, ERNIE-Doc, ERNIE-Sparse on a news text classification task Hyperpartisan (Kiesel et al., 2019). We found that recurrent memory augmentation improves scores for all base models, and even achieves SOTA for the task by outperforming other alternatives for long sequences.

---

### Official Review · Reviewer_ien2 · 2022-07-11

**Rating:** 6
**Confidence:** 3
**Soundness:** 3 good
**Presentation:** 3 good
**Contribution:** 3 good

**Summary:**

The author proposed a memory-augemented transformer where special memory tokens are added to the attention mechanism and test its performance on language modeling and sequence modeling tasks. The author shows that their technique requires less memory in the language modeling tasks to achieve the same accuracy comparing to state-of-the-art model and outperformed baselines where processing long sequence is required.

**Questions:**

1. How does the method perform on other LM related downstream tasks?
2. Can the method match the baseline's accuracy and how will that hurt the time and memory requirement for the method?

------------------------------Update------------------------------------

After reading the author's response, I believe that the author addressed my second concern. My first concern was partially addressed and further effects on other downstream tasks maybe of independent research interests. The author added discussions regarding the wider impact of the method. In the experiment section, the author included more details about their experiment settings and hyper-parameter selection, when another reviewer raised concerns. This made the experiments much clearer and therefore I would like to raise my rating.

**Ethics Review Area:**

["I don’t know"]

**Limitations:**

The author may use a separate paragraph to explain further the societal impact of their work as it is related to large language models, which may raise environmental and other concerns.

**Strengths And Weaknesses:**

The method that the author proposed is simple and their experiments showed strong performance of their method. However, the experiments focuses mainly on the copy and reverse tasks and it remains to be seen how the method performs on other tasks. Also, the perplexity that the author shows is slightly worse than that of baseline. It is remained to be shown whether their method can match the perplexity of the baselines and whether that would mean higher training cost (time and memory), etc.

---

> ### Author Response · Authors · 2022-08-01
> **Response to Reviewer ien2**
>
> The main research goal of the paper is to propose and study a recurrent version of the transformer as a general sequence processing model. This is why we selected standard sequence processing benchmarks traditionally adopted in the literature to assess basic properties of new architectures and left extended evaluation on NLP tasks for future work. Nevertheless, we have already started to test RMT on NLP tasks requiring long document processing and added the first results to the updated version of the paper.
>
> Response to Q1
>
> Up to date we have conclusive results for a popular long text classification benchmark Hyperpartisan (Kiesel et al., 2019). We added recurrent memory to the most widely adopted models from HuggingFace Transformers library - BERT-base, RoBERTa-base, DeBERTa-base, T5-base. With the input segment size of 512 including 10 memory tokens and recurrence over 4 segments we were able to significantly improve metrics for the majority of the models. Moreover, a combination of recurrent memory with RoBERTa-base resulted in an achievement of a new state of the art quality by beating metrics published for BigBird, Longformer, Linformer, Graph-roberta, ERNIE-Doc, ERNIE-Sparse. We included these results in the updated version of the paper.
>
> Response to Q2
>
> Indeed, for a language modeling task RMT shows slightly weaker perplexity compared to Transformer XL but at the same time it uses up to 10 times less memory for storage of representations. Also, we were able to improve Transformer-XL perplexity on WikiText-103 by combining it with recurrent memory from RMT: Tr-XL+RMT (150+10) 23.99 < Tr-XL (150) 24.12; Tr-XL+RMT (75+5) 24.47 < Tr-XL (75) 24.68. To get these results we trained Tr-XL+RMT twice longer than Tr-XL. However, Tr-XL alone did not benefit from the longer training.
>
> Limitations
>
> In our viewpoint adoption of recurrent memory augmentation positively affects limitations of large LMs. Our method allows to learn more compact sequence representations and improve existing pretrained models without extensive additional compute, thus making practical machine learning applications more energy efficient and environmentally friendly. We added this discussion to the updated version of the paper.

---

### Meta-Review · Area_Chair_3Xf6 · 2022-08-30

**Recommendation:** Accept
**Confidence:** Less certain

**Metareview:**

The paper proposes a memory augmented architecture for transformers to deal with the segment-based long sequences. The idea is simple and easy to follow. A special memory token is added to the transformer and corresponding memory operations are introduced to control the storage of the information from previous segments. The experiments show the comparison between the proposed method and the Transfomer-XL architecture. The reviews are overall positive.

**Award:**

No

---

### Decision · Program_Chairs · 2022-09-14

Accept